# PROBING IN THE DARK:
# MAXIMUM STATE ENTROPY IN POMDPS

**Yonatan Ashlag, Mirco Mutti, Aviv Tamar, Kfir Y. Levy**
Department of Electrical and Computer Engineering, Technion - Israel Institute of Technology
`yonatan.ashlag@campus.technion.ac.il`
`{mirco.m,avivt,kfirylevy}@technion.ac.il`

## ABSTRACT

Sample efficiency is one of the main bottlenecks for optimal decision making via reinforcement learning. Pretraining a policy to maximize the entropy of the state visitation can substantially speedup reinforcement learning of downstream tasks. It is still an open question how to maximize the state entropy in POMDPs, where the true states of the environment, or their entropy, are not observed. In this work, we propose to maximize the entropy of a sufficient statistic of the history, which is called an information state. First, we show that a recursive latent model that predicts future observations is an information state in this setting. Then, we provide a practical algorithm, called LatEnt, to simultaneously learn the latent model and a latent-based policy maximizing the corresponding entropy objective from reward-free interactions with the POMDP. We empirically show that our approach induces higher state entropy than existing methods, which translates to better performance on downstream tasks. As a byproduct, we open-source PROBE, the first benchmark to test reward-free pretraining in POMDPs.

## 1 INTRODUCTION

Although Reinforcement Learning (RL) has been essential to the recent advancements of AI, most notably as a technique to align language models with users' preference (Christiano et al., 2017) or to train reasoning models on top of them (Guo et al., 2025), the widespread adoption of its powerful paradigm to a variety of other domains and applications is far from being trivial.

One issue when applying RL is to make sure that the initial decision model, henceforth called a *policy*, is able to explore the environment and collect relevant feedback to update itself in the direction of the desired objective. When this applies, e.g., because we can initialize the learning process with sophisticated foundation models (Brown et al., 2020) or a random initialization is enough to collect informative feedback (Laidlaw et al., 2023), RL has been mostly successful. Hence, the need for general techniques to pre-train a policy for RL arises naturally.

Hazan et al. (2019) have proposed to incentivize the policy with the entropy of the visitation distribution it induces over the states of the environment, which is called the maximum state entropy objective. Under ideal conditions, the resulting policy provides a worst-case optimal initialization to an unknown RL task (Eysenbach et al., 2022): By inducing a uniform state distribution it maximizes the probability of collecting the feedback that is relevant to the task, which is typically a function of the state in RL. While the ideal conditions are unrealistic, a multitude of works have empirically demonstrated the power of state entropy pretraining in a variety of more realistic settings (Mutti et al., 2021; Liu & Abbeel, 2021b; Seo et al., 2021; Yarats et al., 2021, and others).

However, how to translate maximum state entropy maximization in settings with partial observability, which is ubiquitous in real-world domains, is much less understood. How can we maximize the state entropy if we do not get to observe the states in the first place? Previous works (Seo et al., 2021; Yarats et al., 2021; Zamboni et al., 2024b;a) addressing state entropy pretraining in POMDPs (Åström, 1965) have mostly turned to a naïve extensions of the fully observable approach by maximizing the entropy of the partial observations instead of the states. These have had some success in POMDPs with slight partial observability, such as the type that can be easily overcome by stacking a few subsequent observations (e.g., Tassa et al., 2018). Unfortunately, these approaches

are bound to fail on POMDPs with more "general" observability. Zamboni et al. (2024b) formally characterize those settings through the properties of the emission function of the POMDP, which is described by a matrix $O$ when the POMDP is discrete. Their Theorem 4.1 demonstrates that maximizing the observation entropy is enough whenever the maximum singular value $\sigma_{\max}(O)$ of $O$ and the one $\sigma_{\max}(O^{\circ-1})$ of its Hadamard inverse $O^{\circ-1}$ (where $O_{ij}^{\circ-1} = 1/O_{ij}$) are both small. Coarsely, $\sigma_{\max}(O)$ is large when a state can emit various distinct observations, while $\sigma_{\max}(O^{\circ-1})$ is large when a single observation can be emitted by several distinct states. To the best of our knowledge, the literature does not provide a solution for those settings.

In this work, we aim to fill this important gap by developing surrogate objectives and methodologies to address state entropy pretraining in general POMDPs. This effort required contributions on various dimensions—theoretical, methodological, empirical—which we summarize below.

- As a preparatory step, we analyze what constitutes a sufficient statistic of the history of observations—a.k.a. *Information State* (IS)—to maximize the state entropy objective in POMDPs, extending previous results for reward maximization (Subramanian et al., 2022).

- Next, our main contribution is to design a surrogate of the state entropy objective that we can entirely estimate from observations. The idea is to maximize the entropy of a properly designed IS-like statistic instead of the state. To this end, we define a *predictive latent* as a compact statistic that is sufficient to predict future observations. We propose a corresponding *maximum latent entropy objective* to pre-train a policy for general POMDPs.

- We provide a practical algorithm, called *LatEnt*, which is able to simultaneously learn a compact, approximate predictive latent model and a policy maximizing the corresponding latent entropy in challenging POMDPs with continuous states and actions.

- Further, we provide *PROBE*, a continuous control benchmark to evaluate pretraining in POMDPs encompassing observability challenges that are not present in literature domains.

- Using PROBE, we evaluate LatEnt against maximizing the entropy of the true states with oracle access and against maximizing the entropy of the observations. In terms of induced state entropy, LatEnt thoroughly outperforms the latter while falling close to the former, ideal, ceiling. The pre-trained policy allows PPO (Schulman et al., 2017) to learn tasks that are out-of-reach for PPO from scratch and DreamerV3 (Hafner et al., 2023).

Our work provides a substantial advancement over the state of the art in pretraining policies for RL in challenging POMDPs. The implementation of both LatEnt and PROBE are open-sourced at `https://github.com/JonathanAshlag/LatEnt` to spark further research on this important problem.

## 2 PRELIMINARIES

**Notation.** In the following, we denote $[N] := \{1, \dots N\}$ for a constant $N \in \mathbb{N}$. We denote $\Delta_{\mathcal{X}}$ a probability measure on the space $\mathcal{X}$ and $f : \mathcal{X} \to \mathcal{Y}$ a function from a space $\mathcal{X}$ to $\mathcal{Y}$. We denote $\mathcal{X}^t$ as the product space of $t$ copies of $\mathcal{X}$. We denote a sequence $x_1, \dots x_t$ as $x_{1:t}$.

**POMDP.** In this paper, we consider the problem of sequential decision making under partial observability, which is typically modeled with a partially observable Markov decision process (POMDP, Åström, 1965). The POMDP is defined through the tuple $(\mathcal{S}, \mathcal{A}, \mathcal{O}, O, P, T)$, in which $\mathcal{S}$ is a space of unobserved states, $\mathcal{A}$ is a space of actions, $\mathcal{O}$ is a space of observations, $O : \mathcal{S} \to \Delta_{\mathcal{O}}$ is an emission function, $P : \mathcal{S} \times \mathcal{A} \to \Delta_{\mathcal{S}}$ is a transition model, and $T \in \mathbb{N}$ is a decision horizon. An episode of interaction goes as follows. An initial state is drawn as $s_1 \sim P_{init}(\cdot)$, where $P_{init}(\cdot)$ denotes the initial state distribution. For every step $t \in [T]$, the agent observes $o_t \sim O(\cdot|s_t)$ and takes an action $a_t \in \mathcal{A}$, while the state of the process transitions according to $s_{t+1} \sim P(\cdot|s_t, a_t)$. The process stops when the final state $s_T$ is reached and $o_T$ is emitted.

**Objective.** The decision task is typically described by a reward function $R : \mathcal{S} \times \mathcal{A} \to \mathbb{R}$, which assigns a numerical value $r_t = R(s_t, a_t)$ to all the decisions taken within an episode. The aim of the agent is to maximize the expected cumulative rewards $\mathbb{E}[\sum_{t \in [T]} r_t]$ through their actions.[1]

---

[1] Here we do not include the reward function in the POMDP definition, since we will consider different rewards for the same POMDP, representing different tasks, and reward-free settings as well.

**Policy.** The strategy to which the agent select their action is governed by a *policy*. In the most general terms, a policy $\pi$ maps an history $h_t = (o_{1:t}, a_{1:t-1})$ to a distribution over the action to take at step $t$. Formally, $\pi := (\pi_t : \mathcal{H}_t \to \Delta_{\mathcal{A}})_{t \in [T]}$, where $\mathcal{H}_t = \mathcal{O}^t \times \mathcal{A}^{t-1}$ denotes the space of $t$-steps histories. We denote by $\Pi$ the space of history-based policies of this kind.

**Information state.** The space of histories $\mathcal{H}_t$ grows exponentially in $t$, which makes a full representation of an history-based policy intractable. To circumvent the issue, one can condition the policy $\pi_t$ on a compact representation of the history $z_t = \sigma_t(h_t)$, for some mapping $\sigma_t$, containing all the information that is needed to identify the optimal action. This compact representation has been called an *information state* (Subramanian et al., 2022), which is defined as follows.

**Definition 1** (Information state for reward POMDP). *For every $t$-step history $h_T \in \mathcal{H}_T$, a sequence of statistics $(z_t = \sigma_t(h_t))_{t \in [T]}$ is an* information state process *if it holds*

IS1) ***Sufficient for reward prediction***, *i.e.,* $\mathbb{E}[r_t \mid h_t, a_t] = \mathbb{E}[r_t \mid z_t = \sigma_t(h_t), a_t] \; \forall a_t$
IS2) ***Sufficient to predict itself***, *i.e.,* $\mathbb{P}[z_{t+1} \mid h_t, a_t] = \mathbb{P}[z_{t+1} \mid z_t = \sigma_t(h_t), a_t] \; \forall a_t$

*where the condition* IS2 *is also implied by the stronger conditions*

IS2a) ***Evolves recursively***, *i.e., there exist* $(\phi_t)_{t \in [T]}$ *such that* $z_{t+1} = \phi_t(o_{t+1}, z_t, a_t) \; \forall t \in [T-1]$
IS2b) ***Sufficient to predict observations***, *i.e.,* $\mathbb{P}[o_{t+1} \mid h_t, a_t] = \mathbb{P}[o_{t+1} \mid z_t = \sigma_t(h_t), a_t] \; \forall a_t$

The most common information state is the *belief* (Kaelbling et al., 1998). The belief $b_t \in \Delta_{\mathcal{S}}$ encodes the probability of the underlying state given the history $h_t$. Starting from a prior distribution $b_1$, which is typically uniform, the belief is updated as

$$b_t(s|x_t, a_{t-1}, b_{t-1}) \propto O(x_t|s) \sum_{s' \in \mathcal{S}} P(s|s', a_{t-1}) b_{t-1}(s'). \tag{1}$$

## 3 PRETRAINING OBJECTIVES FOR POMDPS

In the fully observable setting—in which the observation emitted from a state always coincides with the state itself—Hazan et al. (2019) have introduced a reward-free pretraining objective of the form

$$\textit{Maximum state entropy objective:} \quad \max_{\pi \in \Pi} H(d^{\pi}(s)), \tag{2}$$

where $H(d^{\pi}(s)) = \sum_{s \in \mathcal{S}} d^{\pi}(s) \log d^{\pi}(s)$ is the entropy and $d^{\pi}(s) = \sum_{t \in [T]} \mathbb{P}(s_t = s|\pi)/T$ is the state distribution induced by the policy $\pi$ over the states. The promise of objective 2 is that a maximum state entropy policy will cover all the states of the environment with high probability, ideally serving as an optimal initialization for any downstream reward maximization task.

On the footsteps of prior works (Seo et al., 2021; Yarats et al., 2021; Zamboni et al., 2024b;a), here we are interested in the problem of state entropy maximization (objective 2) in POMDPs. How can we maximize the entropy when our decision policy can only get a history of observations? First, we aim to establish what is an *information state* for the state entropy objective. Currently, the theory of IS is limited to the reward maximization problem (Subramanian et al., 2022). It is not obvious that the properties in Definition 1 would make for an IS for the state entropy objective, since the latter belongs to a class of problems that generalizes reward maximization, the objective class extending from linear to convex functions of $d^{\pi}(s)$ (Hazan et al., 2019). We call this kind of problems a POMDP with a convex objective, such that the agent aims to maximize $F(d^{\pi}(s))$. We establish the properties of an IS for POMDPs with convex objectives as follows.

**Theorem 1.** *An information state as of Definition 1 is also an information state for a POMDP with a convex objective $F(d^{\pi}(s))$ if condition* IS1 *holds for any function $R : \mathcal{S} \times \mathcal{A} \to \mathbb{R}$ simultaneously.*

*Proof sketch.* The proof (see Appendix B) is based on a result of Hazan et al. (2019) showing that a convex objective $F(d^{\pi}(s))$ can be equivalently optimized by solving a sequence of reward maximization problems and combining their optimal policies into an appropriate mixture. In our setting, each problem in the sequence requires to solve a reward POMDP, for which we know how to characterize an IS (Definition 1). All we need is to enforce that each of the policies in the sequence is based on a common IS, so that we can build their mixture. This is obtained by choosing an IS such that IS1 holds for any reward in the sequence, while IS2 (or IS2a, IS2b) transfers trivially. □

Theorem 1 gives a formal characterization of an IS, but does a statistic of this kind even exist? The history of observations trivially fulfills the properties. Another is the belief (Kaelbling et al., 1998). This implies that one can address objective 2 in POMDPs by learning a policy from beliefs to actions with any methodology for state entropy maximization, a route that has been considered in Zamboni et al. (2024a). There are two caveats. Computing the belief, which evolves as in equation 1, requires knowledge of the transition model $P$ and emission $O$. Moreover, we need to access the true state of the POMDP to estimate $H(d^\pi(s))$. These two conditions may apply when we learn in simulation, but they are unrealistic for real-world POMDPs. Here we consider a general setting as follows:

**Assumption 1.** *$P, O$ are not known and we can access the state of the POMDP at most $O(1)$ times.*

The assumption above encompasses a variety of settings in which the true state is either never accessed or only scarcely, e.g., by installing additional cameras capturing exocentric views of a robot that is only equipped with egocentric views in the nominal operation (Levine et al., 2016). In the setting where a constant access to the true state is available, the small batch of true state information may be used for model selection or hyper-parameter tuning but cannot serve as the learning objective, which require polynomial access to the true states in general, as proved by Lee et al. (2023). Thus, we search for a learnable surrogate objective that we can entirely estimate from observations.

### 3.1 THE MAXIMUM LATENT ENTROPY OBJECTIVE

While Assumption 1 rules out the possibility to maximize the state entropy directly, the obvious alternative is to maximize the entropy of an IS that is sufficient to predict *any* reward function (like in the statement of Theorem 1). Maximizing the surrogate objective would coarsely fulfill the same promise of the original one: Just like an ideal state entropy policy gives a worst-case optimal initialization for an unknown task, as it allows to sample the reward function uniformly over the states, an ideal policy for the IS entropy allows to sample the reward uniformly over the information states. In POMDP tasks, learning an optimal policy mapping information states to actions is the best we can hope for (Subramanian et al., 2022).

There are complications, however. We only have sufficient conditions for an IS. Fulfilling these conditions is easy, e.g., by retaining the whole history as an IS, but the complexity of the resulting entropy problem depends on the size of the space of information states we designed. Think about maximizing the entropy of the space of histories $\mathcal{H}$: The pretrained policy would allow for sampling the downstream reward uniformly over histories. However, the sample complexity for estimating the reward function uniformly well over the histories grows with the size of the corresponding space, i.e., exponentially $|\mathcal{H}| = |\mathcal{O}|^T |\mathcal{A}|^{T-1}$.

Thus, we aim to design the surrogate objective on a statistic that is both sufficient to estimate, and eventually maximize, any downstream reward—like the history—but it is also compact. We look for a statistic with the information state properties (see Definition 1 and Theorem 1) that we can learn entirely from observations. Since we are in a reward-free setting, feedback about the IS1 cannot be gathered from the POMDP. Thus, the statistic focuses on the prediction of future observations.

**Definition 2** (Predictive latent). *Let $L : \mathcal{H} \to \mathcal{L}$ a model to map histories to a latent space $\mathcal{L} \subseteq \mathbb{R}^d$. We call $\ell = L(h)$ a predictive latent if IS2a and IS2b hold for every $h \in \mathcal{H}$, where $l$ takes the place of $z$ and $L$ incorporates the functions $\sigma_t$ and $\phi_t$ of the conditions IS2a and IS2b.*

We can finally design the surrogate objective on top of a predictive latent as

$$\text{Maximum latent entropy objective: } \max_{\pi \in \Pi} H(d_L^\pi(\ell)), \qquad (3)$$

where $d_L^\pi(\ell) := \sum_{t \in [T]} \mathbb{P}(l_t = l | \pi, L)/T$ is the distribution over latent states $\ell \in \mathcal{L}$ induced by a policy $\pi$ under the model $L$. In our experiments, we will show how the introduced objective 3 positively correlates with the original state entropy objective 2 and how it allows to pre-train effective policies for a variety of downstream tasks in challenging POMDPs. We now compare the latent entropy objective with the most common approach of maximizing the entropy of observations (Seo et al., 2021; Zamboni et al., 2024b):

$$\text{Maximum observation entropy objective: } \max_{\pi \in \Pi} H(d^\pi(o)), \qquad (4)$$

where $d^\pi(o)$ is the distribution over observations induced by $\pi$. Especially, we can prove that a policy pre-trained with the latent entropy objective can be adapted to a larger set of downstream tasks than an observation-based policy, which is sufficient to maximize 4 (Hazan et al., 2019).

**Theorem 2.** *A predictive latent fulfills* IS1 *for a larger class of rewards than the observation.*

In Appendix B, we provide a proof of the theorem and an illustrative POMDP example. In Section 5, we will empirically compare objectives 3 and 4, showing that maximum latent entropy provides a much more effective policy pretraining than the observation entropy in challenging POMDPs.

In the next section, we will detail how to learn a predictive latent model from observations in practice. While there are many different model designs (e.g., Hafner et al., 2019a; Zintgraf et al., 2019) that satisfy Definition 2, we hypothesize that a crucial aspect is to keep the model "compact", so that the latent keeps all the information needed to predict future observations but ideally nothing more, focusing the maximum entropy policy on exploring through dynamically different latent states.

## 4    THE LATENT ALGORITHM

We present LatEnt, our approach for learning maximum latent entropy policies. LatEnt optimizes objective 3 by simultaneously learning a latent model and a policy that maximizes entropy in the learned latent space. We describe below the key components of LatEnt: The latent representation learning (Sec 4.1), the policy structure (Sec 4.2), and the entropy estimation method (Sec 4.3).

### 4.1    LATENT DYNAMICS LEARNING

The predictive latent model is implemented as a latent dynamics model (Hafner et al., 2019b). At each step $t$, the observation $o_t$ and previous action $a_{t-1}$ are embedded through separate MLP encoders. A recurrent neural network processes these embeddings sequentially and outputs the latent state $\ell_t = f_\theta(o_t, a_{t-1}, \ell_{t-1})$. For the initial timestep $t = 0$ we set $a_{-1} = 0$ as a null action input.

Our training objective combines observation prediction with latent space regularization to encourage the latent to be both predictive and compact. The model learns to predict the next observation $\hat{o}_{t+1} = p_\theta(\ell_t, a_t)$ given the current latent state and action, which, combined with the recurrent structure, means that $\ell_t$ contains the information to predict $\ell_{t+1}$ (Ni et al., 2024).

To explicitly encourage the latent representation to only retain the information needed for prediction, we employ a dual regularization scheme inspired by KL balancing (Hafner et al., 2020). We train an auxiliary decoder to predict the next latent state $\hat{\ell}_{t+1} = g_\theta(\ell_t, a_t)$ and we encourage the actual next latent $\ell_{t+1}$ to stay close to this prediction. This regularization naturally favors a compact representation: Components of $\ell_{t+1}$ that are not predictable from $(\ell_t, a_t)$ are penalized biasing the model to discard unnecessary information. The full training objective is:

$$\min_\theta \mathcal{L}(\theta) = \sum_{i=1}^{T} (p_\theta(\ell_t, a_t) - o_{t+1})^2 + \alpha(g_\theta(\ell_t, a_t) - sg(\ell_{t+1}))^2 + \beta(\ell_{t+1} - sg(g_\theta(\ell_t, a_t)))^2 \quad (5)$$

where $sg(\cdot)$ denotes stop-gradient. While we chose to learn the dynamics via maximum likelihood, note that our method and analysis can be extended to contrastive learning (Choshen & Tamar, 2023). Note that learning $p_\theta(\ell_t, a_t)$ by minimizing the MSE implicitly assumes that the observation given the latent is distributed as a Gaussian. This is a good fit for continuous control experiments, like those considered in this paper. More expressive distribution can be considered with little change to the algorithm, by simply switching to a Variational Auto Encoder (VAE, Kingma & Welling, 2014).

### 4.2    POLICY DESIGN

Hazan et al. (2019) have demonstrated that Markovian stochastic policies are sufficient to maximize the state entropy, which means that a policy conditioned on the latent state is sufficient for the maximum latent entropy. Thus, we use the latent states $\ell_t$ directly as input to a feedforward policy network without any additional recurrent structure. Prediction-based representations have also been shown to provide valuable inductive biases that aid generalization (Dosovitskiy & Koltun, 2016). Empirically, we observe that augmenting the policy input with the current observation improves the performance, especially during initial training stages where the latent model changes rapidly.

---

**Algorithm 1** LatEnt: On Policy RL Pretraining

---

**Require:** `encoder_update_ratio`, `n_warmup`, batch size $N$
 1: Initialize latent dynamics model parameters $L_\theta$ and agent parameters $\pi_\psi$
 2: Initialize replay buffer $\mathcal{D} \leftarrow \emptyset$
 3: Collect `n_warmup` trajectories with a uniform-actions policy and pre-train $L_\theta$
 4: **for** iteration $= 1, 2, \ldots,$ `training_horizon` **do**
 5:     Collect $N$ trajectories $\{\tau_i\}_{i=1}^N$, where $\tau_i = \{(a_{t-1}, o_t, \ell_t)\}_{t=0}^T$ using $\pi_\psi$ and $L_\theta$
 6:     $\mathcal{D} \leftarrow \mathcal{D} \cup \{\tau_i\}_{i=1}^N$
 7:     Compute latent-entropy rewards $\{\tau_i^{\text{rewards}}\}_{i=1}^N$, where $\tau_i^{\text{rewards}} = \{r_{i,t}\}_{t=0}^T$, using the latent on policy batch $\{\ell_{i,0:T}\}_{i=1}^N$ as in equation 6
 8:     $\{\tau_i^{\text{RL}}\}_{i=1}^N = \{\tau_i \cup \tau_i^{\text{rewards}}\}_{i=1}^N$
 9:     Update $\pi_\psi$ with PPO using $\{\tau_i\}_{i=1}^N$
10:     **if** iteration mod `encoder_update_ratio` $= 0$ **then**
11:         Train $\theta$ on $\mathcal{D}$ to minimize equation 5
12:     **end if**
13: **end for**

---

### 4.3 NEAREST NEIGHBOR ENTROPY MAXIMIZATION

Entropy estimation in high-dimensional spaces is a challenging problem. A popular approach in the RL community (Mutti et al., 2021; Yarats et al., 2021; Seo et al., 2021) is to use a non-parametric entropy estimator (Singh et al., 2003).

Given samples $\{z_i\}_{i=1}^N$, the estimator takes the form $\hat{H}_N^k(Z) \propto \sum_{i=1}^N \log \|z_i - z_i^{k\text{-NN}}\|_2$, where $z_i^{k\text{-NN}}$ is the k-nearest neighbor of $z_i$ within the batch. The latter is proved to be asymptotically unbiased and consistent (Singh et al., 2003). In practice, the estimated entropy an intrinsic reward (Liu & Abbeel, 2021b)

$$r^i(z_i) := \log(||z - z_i^{k\text{-NN}}||_2 + c) \tag{6}$$

$c \in \mathbb{R}$ Is a constant added to preserve numerical stability. For more details see Appendix D.2.

### 4.4 LATENT ALGORITHM

The complete LatEnt algorithm, for which the pseudocode is in Algorithm 1, combines the three components described above. Specifically, we learn a latent dynamics model as detailed in Section 4.1 and we build the policy on top of these latent representations as described in Section 4.2.

The main challenge lies in balancing the synergy between these components. The latent model has to be up-to-date to provide a meaningful entropy objective, but it can cause instability by changing too rapidly, since the policy also depends on the latent space. To address this trade-off, we employ a two-stage training approach. First, we warm up the latent model by pretraining it on data collected by a policy initialized to output near-uniform action probabilities. During online training, we update the dynamics model less frequently than the policy to maintain stability in the latent space.

The policy is learned with on-policy trajectories using PPO (Schulman et al., 2017). For each update, we compute the latent entropy rewards (Eq. 6) using the current on-policy batch. Since the entropy is harder to estimate than typical RL objectives (Singh & Póczos, 2016; Ashlag et al., 2025), we use a larger batch size than standard on-policy methods.

## 5 EXPERIMENTS

In this section, we provide an empirical validation of our method. We designed the following experiments to answer three key questions: (Q1) Can LatEnt induce higher state entropy than observation-based objectives? (Q2) Does LatEnt pretraining enable rapid adaptation to downstream tasks? (Q3) Which components of the LatEnt algorithm are most critical for its performance?

## 5.1 PROBE BENCHMARK

To properly evaluate these questions, a suitable benchmark is needed. Indeed, Zamboni et al. (2024b, Theorem 4.1) show that maximizing the observation entropy is enough whenever the maximum singular values $\sigma_{\max}(O)$ of $O$ and $\sigma_{\max}(O^{\circ-1})$ of the Hadamard inverse (having elements $O_{ij}^{\circ-1} = 1/O_{ij}$) are both small. Here we are interested in settings in which either $\sigma_{\max}(O)$ or $\sigma_{\max}(O^{\circ-1})$ is large, which coarsely means that a state can emit various distinct observations or a single observation can be emitted by several distinct states.

A popular benchmark in prior work on unsupervised exploration in POMDPs (Seo et al., 2021; Yarats et al., 2021) is the DeepMind Control Suite (Tassa et al., 2018). In the latter, $\sigma_{\max}(O^{\circ-1})$ and $\sigma_{\max}(O)$ are both relatively small in all the domains. The former is small because every state can emit just one observation. The latter is not small *per se*, as every observation can be emitted by several distinct states, e.g., same position with different velocities, but it becomes small once you stack a few consecutive observations (frames), which is the default setup of prior works.[2] Other POMDP benchmarks either focus on easy exploration scenarios that primarily test memorization capabilities (Morad et al., 2023), or scenarios in which the challenges of exploration and partial observability are orthogonal (Hausknecht & Stone, 2015; Pasukonis et al., 2022).

To address this gap in POMDP benchmarks, we introduce **PROBE** (PaRtially OBservable Exploration benchmark), a suite of partially observable continuous control environments to test both unsupervised exploration at pretraining and downstream RL, requiring agents to explore despite misleading or incomplete observations. Visualizations of PROBE are in Figure 1.

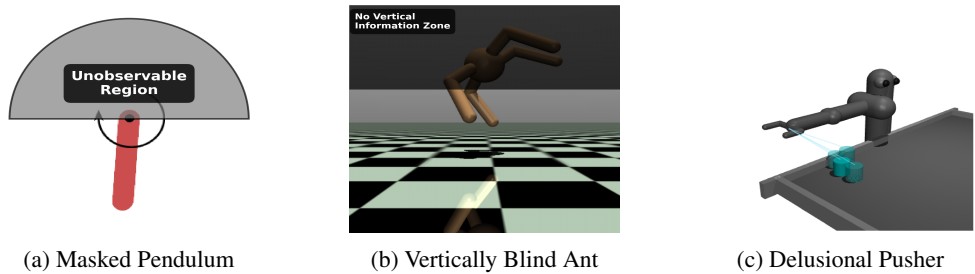

(a) Masked Pendulum      (b) Vertically Blind Ant      (c) Delusional Pusher

Figure 1: **PROBE** (a) Top half is masked and velocity is unobserved (b) Height-related information is masked above a threshold (c) The further the arm the noisier the puck observation becomes.

**Masked Pendulum:** Based on Gymnasium's classic pendulum environment (Towers et al., 2024). Partial observability is introduced by masking the top half of the environment—when the pendulum is above $y = 0$ is unobservable—and by occluding velocity from the observation vector. Further, we fix the pendulum's starting position to the bottom of the circle and cap the maximum torque. While the environment is low-dimensional (3D states, 2D observations, 1D actions), exploring the hidden upper region requires a non-trivial behavior and the right incentive. Many states emit the same observation means that $\sigma_{\max}(O)$ is high, for further analysis see appendix C.1.

**Vertically Blind Ant:** Based on Mujoco's Ant environment. Partial observability is introduced by masking all height-related information when the $z$ coordinate of the Ant's torso exceeds a threshold. Coarsely, the Ant's z-axis is fully observable only when the Ant is on the ground. Furthermore extrinsic forces are unobserved at all times. This induces challenges that are similar to the Masked Pendulum in nature, but on a larger scale (105D states, 27D observations, 8D actions). Many states emit the same observation means that $\sigma_{\max}(O)$ is high.

**Delusional Pusher:** Based on MuJoCo's Pusher environment (20D states, 20D observations, 7D actions). The partial observability is obtained by adding 3D Gaussian noise to the observation of the puck location. This makes the $\sigma_{\max}(O^{\circ-1})$ increase. The further the end effector is from the puck, the noisier the puck position becomes, simulating noisy sensor readings. This affect makes $\sigma_{\max}(O)$ increase slightly. This double effect of the noise makes the pusher domain the hardest in the benchmark. Given that most of the reasonable tasks in this domain are related to the puck, we will focus on entropy maximization of the puck's position (more details below).

---

[2]See Appendix C.4 for further discussion on unsupervised exploration in DeepMind Control Suite.

## 5.2 Pretraining with LatEnt

(Q1) Can LatEnt induce higher state entropy than observation-based objectives?
LatEnt outperforms maximum observation entropy in all the PROBE environments.

**Baselines:** We compare LatEnt against baselines optimizing maximum state entropy—with oracle access to the states—and maximum observation entropy, like in Seo et al. (2021); Yarats et al. (2021). To ensure fair comparison, all methods use the same policy architecture as LatEnt, with policies built on top of learned latent dynamics models (see Section 4.2). For maximum observation entropy, we follow the same training procedure as LatEnt but calculate the entropy rewards on the observation vectors instead of latent states. For maximum state entropy, we use the same approach but calculate entropy on the true states, serving as an ideal ceiling for the evaluation of others. To focus only on the puck in Delusional Pusher, we implement LatEnt to only predict the puck's position, whereas for the baselines we calculate the entropy over the puck's observed/true features only (see Apx. D).

**Results:** Figure 2 shows that LatEnt leads to more state entropy than maximum observation entropy across all environments. This highlights the fundamental mismatch between observation and state entropy in challenging POMDPs. In Delusional Pusher, maximum observation entropy exploits the noise in the emission of observations by moving the arm away from the puck instead of manipulating it, resulting in zero state entropy (Figure 3). In the other environments, the maximum observation entropy policy is confined to the observable dimensions of the state, while LatEnt's dynamics model infers these hidden dimensions from temporal patterns and incentivizes exploring them.

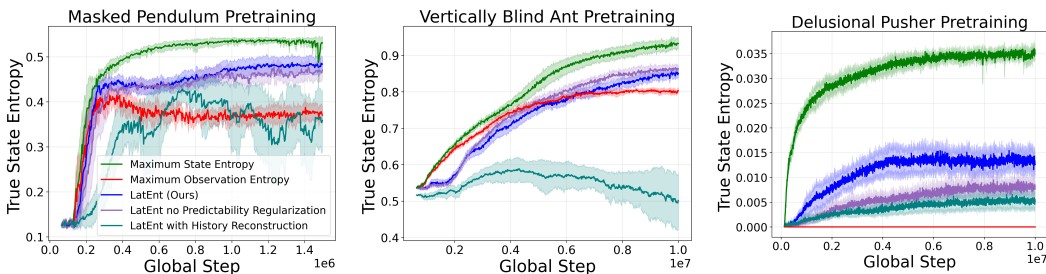

Figure 2: True state entropy is calculated via k-NN estimation on the true (unobserved) states collected by each method during training. The last two lines are variants of LatEnt discussed in the ablations (Sec. 5.4). Lines show mean performance with 95% confidence intervals across 10 seeds.

## 5.3 Finetuning LatEnt Policies to Downstream Tasks

(Q2) Does LatEnt pretraining enable rapid adaptation to downstream tasks?
PPO initialized with a LatEnt policy substantially outperforms learning from scratch

**Downstream tasks:** We design various tasks for each PROBE environment that correspond to high-level "skills", such as navigation, jumping, balancing. All tasks are reward-sparse and require to execute complex maneuvers for which unobserved state information is key (see Apx. C.3).

**Finetuning scheme:** We employ standard PPO with an initial policy coming from the LatEnt pre-training and a randomly initialized critic. The algorithm interacts with the environment online to maximize the task reward. During fine-tuning, the latent dynamics model and policy are treated as one component and optimized end-to-end. This minimal finetuning scheme is designed to isolate the contribution of the policy pretraining from other sophisticated mechanisms that may be used to further improve the performance of our approach—or the baselines—such as transferring previously collected samples or latent-based exploration bonuses.

**Baselines:** Using the same finetuning scheme, we compare a LatEnt initialization against pretraining with maximum observation entropy and maximum state entropy pretraining (recall the latter uses oracle access to true states). We additionally compare against PPO trained from scratch and DreamerV3 (Hafner et al., 2023), the state-of-the-art model-based algorithm for POMDPs.

**Results:** In Figure 4, we show that LatEnt pretraining allows PPO to learn all the tasks, achieving a performance that approaches the maximum state entropy pretraining (which relies on state informa-

Figure 3: Policy rollout snapshots. Left: LatEnt jumps high in the unobserved region while the baselines are confined to the bottom. Right: LatEnt manages to manipulate the puck while the baselines are limited to jitters (random) or move away from the puck to gather noise (observation).

tion at pretraining). As it is evident from the non-zero performance of the initial step, the pretrained policy is able to collect some reward in all the tasks, despite the latter being sparse. Instead, maximum observation entropy performs poorly, solving only 1/6 tasks within the given budget of steps. In the Delusional Pusher tasks, it fails to discover rewards entirely. This supports our claim that observation-based objectives are ill-suited for pretraining in challenging POMDPs. As a testament of the inherent hardness of the considered tasks, both PPO from scratch and DreamerV3 fail to consistently collect rewards in most tasks, further highlighting the benefit of effective pretraining.

## 5.4 ABLATION STUDIES

(Q3) Which components of the LatEnt algorithm are most critical for its performance? Predictive latent and predictability regularization are essential to achieve high state entropy

We empirically validate the design of our latent model with alternatives: (1) LatEnt + history encoding, where the latent model is additionally trained to reconstruct past history—a popular approach for learning in POMDPs (Zintgraf et al., 2019)—(2) LatEnt without predictability regularization.

Empirically, the correlation between the entropy of the history embeddings and the entropy over true states is inferior than standard LatEnt. We hypothesize this is due to the size of the latent space being less compact, as it grows exponentially with the horizon. This conclusion is supported by Fig. 2, in which environments with longer horizon show larger performance gaps between the methods.[3]

Predictability regularization proves beneficial in practice. It provides minimal benefit in the masked environments, where little redundant information exists, but proves crucial in Delusional Pusher where LatEnt without predictability regularization achieves significantly lower state entropy.

## 6 DISCUSSION AND CONCLUSION

Before concluding, we discuss related works (see Apx. A for a thorough overview of the literature).

**Representation Learning in POMDPs.** A crucial component of LatEnt is to learn a predictive and compact latent representation of the history. Latent dynamics models are not new in the literature. One of the most common approaches is a recurrent state space model like Dreamer (Hafner et al., 2019a). Dreamer is similar to our latent model in their training loss, but we turn to a deterministic model to avoid artificially inflating the latent entropy through the model stochasticity. Another approach is to learn a representation that reconstruct the entire history (Zintgraf et al., 2019), which is similar in spirit to our LatEnt + history encoding. Given the poor performance in our ablation study, we hypothesize that the entropy of the latent space is not the right incentive for those approaches, for which state novelty has been more successful (Zintgraf et al., 2021). However, Zintgraf et al. (2021) do not provide a method to extract a policy to reproduce exploration in downstream tasks.

**Maximum State Entropy.** Several previous works have taken on the maximum state entropy objective, introduced by Hazan et al. (2019), to develop practical algorithms for challenging settings,

---

[3]Episode lengths: Pusher: 100 steps, Pendulum: 200 steps, Ant: 1000 steps.

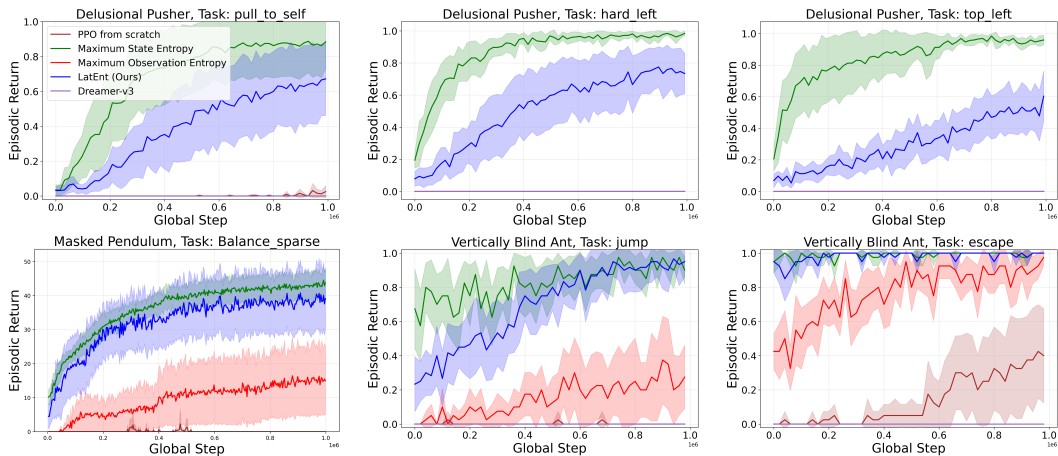

Figure 4: Finetuning on PROBE downstream tasks. All tasks have a training budget of a million steps. For each algorithm, we report the mean and 95% confidence intervals across 10 seeds. Note that methods achieving zero return may overlap visually in the plots.

like continuous control from proprioceptive states (Mutti et al., 2021) and images (Liu & Abbeel, 2021b; Seo et al., 2021; Yarats et al., 2021). Our method is built upon them with some implementation differences. It computes the policy updates fully on-policy, like (Mutti et al., 2021), but maximizes entropy rewards like the others, which use off-policy methods instead of our PPO base.

**Maximum Entropy in POMDPs.** Previous works addressing policy pretraining in POMDPs have mostly turned to a maximum observation entropy approach (Seo et al., 2021; Yarats et al., 2021). Zamboni et al. (2024b) analyze the theoretical limitations of this approach, while we provide an empirical account of their shortcomings. Zamboni et al. (2024a) propose to address these limitations by maximizing the entropy of synthetic states sampled from the sequence of beliefs. This is similar in spirit to our approach of maximizing the entropy of a IS-like statistic. Their method, however, rely on knowledge of $P, O$ and it is not tractable in the domains we consider. Notably, our paper tackles both the open problems left by Zamboni et al. (2024a;a) by providing a scalable algorithm for state entropy maximization in continuous domains, without assuming given $P, O$ or small $\sigma_{\max}(O)$ and $\sigma_{\max}(O^{\circ-1})$.

**Conclusions.** We tackle the important problem of unsupervised pretraining in POMDPs. We developed a novel, scalable algorithm to translate the maximum state entropy framework to the partially observable setting. We show that our algorithm empirically outperforms existing alternatives, providing an important step towards real-world deployment of maximum entropy pretraining in a variety of applications.

## ACKNOWLEDGMENTS

KYL and YA were partially supported by Israel PBC-VATAT, by the Technion Artificial Intelligence Hub (Tech.AI), and by the Israel Science Foundation (grant No. 3109/24). This work received funding from the European Union (ERC, Bayes-RL, Project Number 101041250). Views and opinions expressed are however those of the authors only and do not necessarily reflect those of the European Union or the European Research Council Executive Agency. Neither the European Union nor the granting authority can be held responsible for them.

## ETHICS STATEMENT

This work presents fundamental research in reinforcement learning theory and algorithms. We have carefully reviewed the ICLR Code of Ethics and confirm that our research raises no ethical concerns. The work involves no human subjects or private data collection. All experiments were conducted

in simulated environments. We identify no specific negative societal impacts or deviations from the ICLR Code of Ethics.

## REPRODUCIBILITY STATEMENT

All experiments were run on two NVIDIA RTX 4090 GPUs. The total wall-clock time to re-run all experiments is approximately two weeks. Our proposed algorithm LatEnt is detailed in section 4, with full implementation details and hyperparameters included in Appendix E. Our new evaluation benchmark PROBE is introduced in Section 5, with full implementation details available in Appendix C.3. Proofs of all theoretical claims are provided in Appendix B. The source code for both PROBE and LatEnt is available at `https://github.com/JonathanAshlag/LatEnt`.

## LLM DISCLOSURE

We used LLMs in a limited capacity as a writing assistance tool. Specifically, LLMs were employed to help refine the clarity and readability of selected paragraphs throughout the paper. All content has been reviewed and verified by the authors, who take full responsibility for the accuracy and originality of all statements in this paper.

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

## A    OVERVIEW OF RELATED LITERATURE

We summarize below the literature that most relates to our work.

**POMDPs.**    In the last few decades, the study of POMDPs have received notable attention (Kaelbling et al., 1998). Although classical results have established the intractability of solving a POMDP in general (Papadimitriou & Tsitsiklis, 1987; Mundhenk et al., 2000), several works have studied POMDPs under structural properties that makes the problem approachable, both in the planning (Sondik, 1971; 1978; Smallwood & Sondik, 1973; Subramanian et al., 2022; Golowich et al., 2022) and the learning (Krishnamurthy et al., 2016; Jin et al., 2020; Chen et al., 2022; Liu et al., 2022; Zhan et al., 2023; Zhong et al., 2023) settings. On a more practical side, approximate methods have been developed, ranging from traditional dynamic programming approaches (Hauskrecht, 2000; Pineau et al., 2006) to deep reinforcement learning (Hausknecht & Stone, 2015; Igl et al., 2018; Hafner et al., 2019a; Zintgraf et al., 2019; Choshen & Tamar, 2023). Especially, the Dreamer architecture (Hafner et al., 2019a), in which a latent dynamics model is learned to predict the next observation, provides the inspiration for the latent backbone of our approach.

**Maximum state entropy in MDPs.**    In fully observable settings, Hazan et al. (2019) were the first to introduce the problem of learning an exploration policy by maximizing the state distribution entropy. This reward-free objective has been used as a pre-training objective for downstream RL (Mutti & Restelli, 2020; Mutti et al., 2021; Liu & Abbeel, 2021b; Seo et al., 2021; Yarats et al., 2021) as well as a regularization within RL algorithms to foster state coverage (Islam et al., 2019; Seo et al., 2021; Yuan et al., 2022; Kim et al., 2023; Bolland et al., 2024) and to provide robustness to perturbations of rewards and transition dynamics (Ashlag et al., 2025). Other works have studied various aspects of the state entropy objective, including the statistical complexity of the learning problem (Tiapkin et al., 2023), alternative formulations beyond Shannon entropy (Mutti & Restelli, 2020; Zhang et al., 2021; Guo et al., 2021; Mutti et al., 2022a; Yuan et al., 2022; Nedergaard & Cook, 2022), multi-agent settings (Zamboni et al., 2025), and others (Lee et al., 2019; Tarbouriech et al., 2020; Liu & Abbeel, 2021a; Yarats et al., 2022; Mutti et al., 2022b; Mutti, 2023; Yang & Spaan, 2023; Jain et al., 2023; Zisselman et al., 2023; Li et al., 2024; De Santi et al., 2025; De Paola et al., 2025). The works introducing $k$-NN entropy estimation for MDPs (Mutti et al., 2021; Liu & Abbeel, 2021b; Seo et al., 2021; Yarats et al., 2021) have been a crucial inspiration for the implementation of our algorithm, which is fully on-policy like (Mutti et al., 2021) but decompose the entropy estimate in pseudo-rewards like (Liu & Abbeel, 2021b; Seo et al., 2021; Yarats et al., 2021).

**Maximum state entropy in POMDPs.**    Despite the substantial body of work on the state entropy objective in MDPs, how to maximize the entropy of the states by accessing partial observations only is less understood. In rich-observation settings, e.g., when learning from images, it is not uncommon to cast the state entropy problem as a POMDP (Seo et al., 2021; Yarats et al., 2021), although the partial observability is easily overcome by stacking a few frames to be processed into the desired representations. More recently, Zamboni et al. (2024b;a) have tackled the problem in more general POMDPs, showing the potential pitfalls of computing the entropy on observations (Zamboni et al., 2024b) and *hallucinations* coming from the belief (Zamboni et al., 2024a). However, they fell short of providing a practical algorithm that do not require access to the emission function or transition model. In our work, we advance on those foundations to provide a fully sample-based algorithm for state entropy maximization in general POMDPs. Other work worth mentioning is (Savas et al., 2022), maximizing the entropy of the trajectory of observations instead of the (partially observed) states, and (Svidchenko & Shpilman, 2021), which also considers state entropy maximization with a Dreamer-inspired world model (Hafner et al., 2019a).

## B    PROOFS

### B.1    PROOF OF THEOREM 1

We want to prove what constitutes an information state of a POMDP with a convex objective $F(d^\pi(s))$. Especially, we establish a sufficient condition that follows the same properties of Definition 1 in terms of dynamics prediction, i.e., IS2 or, alternatively, IS2a and IS2b, but requires the condition IS1 of reward estimation to hold for any function $R : \mathcal{S} \times \mathcal{A} \to \mathbb{R}$.

For each $t \in [T]$, let $\sigma_t : \mathcal{H}_t \to \mathcal{Z}$ be a map between histories of length $t$ and a statistic as detailed above. We aim to characterize a policy $\pi_z : \mathcal{Z} \to \Delta_\mathcal{A}$ such that $F(d^{\pi_z}) = F(d^{\pi^*})$, where $\pi^*$ is an optimal history-based policy $\pi^* \in \arg\max_{\pi \in \{\mathcal{H} \to \Delta_\mathcal{A}\}} F(d^\pi)$.

We provide a constructive proof through the MaxEnt algorithm (Hazan et al., 2019), which can be used to compute a policy $\pi_z$ with the desired properties. Hazan et al. (2019) show that the convex problem $F(d^\pi(s))$ can be reduced to a sequence of reward maximization problems. Specifically, for the $h$-th problem in the sequence, the reward function takes the form $R_h(s, a) = \nabla F(d^{\pi_{\text{mix},h}}(s))$ where $\pi_{\text{mix},h}$ is computed at the previous iterations, as detailed below.

For each iteration $h$, we compute a policy $\pi_h : \mathcal{Z} \to \Delta_\mathcal{A}$ that maximizes the reward $R_h$. Since, by the theorem statement, the statistics in $\mathcal{Z}$ are information states for any reward function, including $R_h$, we have that $\pi_h$ matches the performance of the optimal history-based policy for the same reward. Next, we combine $\pi_h$ with previous policies in the mixture $\pi_{\text{mix},h}(a|z) = \sum_{j \in [h-1]} \alpha_j \pi_j(a|z)$ for some appropriate coefficients $\alpha_j$ (see Hazan et al. 2019). Now, we invoke Theorem 4.1 of Hazan et al. (2019) to obtain $F(d^{\pi_{\text{mix},h}}) \geq \max_{\pi \in \{\mathcal{H} \to \Delta_\mathcal{A}\}} F(d^\pi) - \epsilon$ for some $\epsilon > 0$. Finally, taking the limit for infinite iterations, we have $\lim_{h \to \infty} F(d^{\pi_{\text{mix},h}}) \geq \max_{\pi \in \{\mathcal{H} \to \Delta_\mathcal{A}\}} F(d^\pi)$, which proves the theorem.

### B.2 Proof of Theorem 2

We prove the theorem by (i) showing that for every reward function for which the observation fulfills IS1 the predictive latent also fulfills IS1 and (ii) showing an instance for which the predictive latent fulfills IS1 but the observation does not.

(i) In general, a reward function for which the immediate observation fulfills IS1, it can be equivalently written as a function $R : \mathcal{O} \times \mathcal{A} \to \mathbb{R}$. If the predictive latent can predict the next observation as well as the history (IS2b), then this also holds for any function of the observation $\mathbb{P}[f(o_{t+1}) \mid h_t, a_t] = \mathbb{P}[f(o_{t+1}) \mid z_t = \sigma_t(h_t), a_t] \, \forall a_t$, including the reward.

(ii) Consider the example below. It is straightforward to see that the history contain the information to decode the state perfectly. Thus, to be able to predict the next observation as well as the history (IS2a and IS2b), the predictive latent also needs to decode the state perfectly, which means reward prediction IS1 is also given for any reward trivially. Instead, we cannot distinguish the state $s_1$ and $s_3$ from the immediate observation, which means the latter does not fulfill IS1 for any reward such that $R(s_1, a) \neq R(s_3, a)$ for some $a \in \mathcal{A}$.

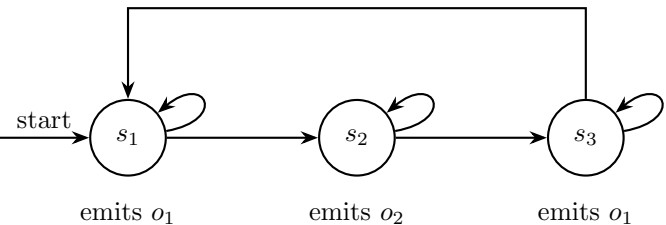

## C PROBE: PArtially OBservable Exploration benchmark

### C.1 Masked Pendulum

We make Gymnasium's classic pendulum environment partially observable by:

- asking the top half of the environment—when the pendulum is above $y = 0$ (observation in that case is [0,0])
- velocity information is occluded at all times
- agent's starting position is fixed to the bottom of the circle.
- maximum torque (action) output is clipped to 0.9 (originally 2).

With these changes, exploring the state space requires learning to gather momentum and understanding the need to cover the unobserved top region. As in the original environment episodic horizon is 200 steps.

To show the effect of our changes on the observability properties of the domains, we discretize the pendulum position and velocities in a grid 16x16 (256 states in total) and compute the value of $\sigma_{\max}(O)$ for each variation.

- The original implementation has $\sigma_{\max}(O) = 1$
- Masking the position in the upper half of the circle $\sigma_{\max}(O) \approx 2.82$
- Masking the position in the upper half of the circle and velocities everywhere $\sigma_{\max}(O) \approx 11.313$

This analysis backs up our design choice to mask both the position and velocities.

**Balance sparse Task:** The agent needs to learn to swing up and balance the pendulum at the (unobservable) summit, reward is:

$$\text{reward} = \begin{cases} 1, & \text{if y} > 0.85, \\ 0, & \text{otherwise.} \end{cases}$$

## C.2 VERTICALLY BLIND ANT

built upon Mujoco's "Ant-v5" we make the environment partially observable by defining a threshold z=0.75 which upon hight related data is masked in the following way:

- z-coordinate of the torso (centre) is shown to be 0.75
- z-orientation of the torso (centre) is masked to 0
- z-coordinate velocity of the torso is masked to 0
- z-coordinate angular velocity of the torso is masked to 0

Furthermore, we exclude center of mass based external forces (78 elements of the observation vector are the "center of mass based external forces on the body parts", Tassa et al. 2018) and the x,y coordinates. For reference when the Ant is standing upright it's hight is unobservable at 1.0. As in the original environment episodic horizon is 1000 steps.

**Jump-sparse Task:** As in Mutti et al. (2021) the ant needs to learn to jump with a sparse reward:

$$\text{reward} = \begin{cases} 1, & \text{if z-coordinate of the torso} > 3.0, \\ 0, & \text{otherwise.} \end{cases}$$

The episode terminates upon task completion, otherwise episode horizon is 1000. Note that the vertical blindness makes this extra difficult as the task is deep in the unobserved region of the POMDP.

**Escape Task** The ant's initial state is upside down, meaning its laying on its back with no velocity. the ant needs to learn to escape this position and return to and upright standing position. reward is:

$$\text{reward} = \begin{cases} 1, & \text{if upright orientation (quaternion-derived body alignment)} > 0.8 \\ & \text{and } 0.5 < \text{z-coordinate of the torso} < 1.0 \\ 0, & \text{otherwise.} \end{cases}$$

## C.3 DELUSIONAL PUSHER

We modify MuJoCo's `Pusher` by corrupting the puck-position observation with distance-proportional Gaussian noise. let $\mathbf{p}_t, \mathbf{e}_t$ denote the puck's and the arm's end effector's 3d coordinates at time t. The agent observes:

- Distance: $d_t = \|\mathbf{e}_t - \mathbf{p}_t\|_2$.
- Noise: $\boldsymbol{\epsilon}_t \sim \mathcal{N}(\mathbf{0}, (\lambda d_t)^2 \mathbf{I}_3)$ (i.i.d. across axes and time).
- Observation: $\tilde{\mathbf{p}}_t = \mathbf{p}_t + \boldsymbol{\epsilon}_t$.

Lastly, as the arm is stationary, we added walls the hight of the puck around the edges of the table to prevent the unwanted behavior of pushing the puck far out of reach.

**Downstream tasks:** we define 3 goal-reaching sparse tasks shown in table 1: All tasks return a positive reward if the pucks true position is within 0.1 of the goal, else 0. In all tasks the hand needs to maneuver the puck a distance more than 3x the distance in the original Pusher task. Furthermore, in contrast to the original Pusher task, in our version the agent does not observe the goal location. puck's origin is set to be (0.65,-0.35).

Table 1: Task variants and geometry

| Task | Goal | Distance from puck's origin |
|------|------|------------------------------|
| Hard left | (0,-0.45) | 0.66 |
| Pull to self | (-0.35 , -0.25) | 1 |
| Top left | (0,0.25) | 0.78 |

## C.4 DEEPMIND CONTROL

To show that introducing the PROBE benchmark is essential for evaluating unsupervised pre-trainign in POMDPs We conducted additional experiments in the most common DeepMind Control suite (Tassa et al. 2018). We summarize in the table below the results obtained by LatEnt against the maximum observation entropy (solution proposed by prior works) and the maximum state entropy (ideal target). As the reader can see, LatEnt and maximum observation entropy achieve nearly identical performance across all metrics. This is in line with the results of Zamboni et al. (2024b), which articulate on the properties for which maximizing the observation entropy is enough.

Table 2: DMC Walker pretraining (10 seeds, mean $\pm$ std)

| | State entropy | Observation entropy | Latent entropy |
|---|---|---|---|
| **Max state entropy** | $2.61 \pm 0.008$ | $1.528 \pm 0.011$ | $0.847 \pm 0.013$ |
| **Max observation entropy** | $2.35 \pm 0.027$ | $1.589 \pm 0.012$ | $0.85 \pm 0.01$ |
| **LatEnt** | $2.36 \pm 0.012$ | $1.58 \pm 0.005$ | $0.863 \pm 0.012$ |
| **Random policy (baseline)** | $2.26 \pm 0.001$ | $1.49 \pm 0.001$ | NA |

Table 3: DMC Hopper pretraining (10 seeds, mean $\pm$ std)

| | State entropy | Observation entropy | Latent entropy |
|---|---|---|---|
| **Max state entropy** | $1.77 \pm 0.056$ | $1.19 \pm 0.039$ | $0.757 \pm 0.042$ |
| **Max observation entropy** | $1.73 \pm 0.03$ | $1.3 \pm 0.01$ | $0.81 \pm 0.01$ |
| **LatEnt** | $1.71 \pm 0.03$ | $1.28 \pm 0.024$ | $0.818 \pm 0.017$ |
| **Random policy (baseline)** | $1.39 \pm 0.002$ | $1.04 \pm 0.001$ | NA |

## D IMPLEMENTATION DETAILS

We hence provide extra implementation details not detailed in the main body of the paper.

### D.1 LATENT IMPLEMENTATION

We built LatEnt on-top of Cleanrl's Huang et al. (2022) continuous action PPO implementation.

**PPO advantage baseline:** While it is common to learn a value function as a baseline and further reduce variance with general advantage estimation, we found value learning in on-policy maximum state entropy too volatile on occasions, as the exact occupancy in each point can shift from batch to batch. On the other-hand the total entropy evolved stably from iteration to iteration so we employed a running average baseline instead.

$$baseline = \text{mean-over-last 20-iterations-per-timestep}$$

$$A_t = \sum_{i=t}^{T} r_i - baseline_i$$

**Truncated Back Propagation Through Time** As part of the dynamics model training, we implemented Truncated Backpropagation Through Time (TBPTT) as suggested by Pasukonis et al. (2022) to help RNNs preserve information over time. We note that we didn't see any significant impact using it, but we kept it in the implementations as it saves memory.

**Entropy maximization of subspaces**: There are scenarios where we want to focus exploration on only a relevant subspace of the state space, as in Delusional Pusher. Since achieving diverse arm positions provides little value for downstream manipulation tasks, we only care about learning to move the puck to diverse locations. LatEnt can be easily modified to direct exploration toward the relevant subspace: the latent dynamics model continues to process full observations, but is trained to predict only future observations of the desired subspace (e.g., the puck's location). The rest of the algorithm remains the same.

Table 4: Hyperparameters used in our experiments.

| Hyperparameter | Masked Pendulum | Vertically blind Ant | Delusional Pusher |
|---|---|---|---|
| Training steps | 1.5e6 | 1e7 | 1e7 |
| Steps per episode | 200 | 1000 | 100 |
| Batch size | 16 | 16 | 32 |
| Entropy reward calculated upon | per episode | full batch | full batch |
| Value of k (k-NN) | 5 | 25 | 25 |
| Action entropy coefficient | 0.001 | 0.002 | 0.005 |
| Policy network hidden size | 512 | 512 | 512 |
| PPO Advantage normalization | False | False | False |
| PPO Clip coefficient | 0.25 | 0.25 | 0.25 |
| PPO update epochs | 4 | 4 | 4 |
| PPO mini batches per epoch | 4 | 4 | 4 |
| Policy learning rate | 3e-4 | 3e-4 | 3e-4 |
| Warm up (batchs collected) | 20 | 40 | 40 |
| Dynamics model (DM) learning rate | 3e-4 | 3e-4 | 3e-4 |
| DM epochs after warm up | 200 | 400 | 200 |
| DM update frequency | $\frac{1}{5}$ | $\frac{1}{5}$ | $\frac{1}{5}$ |
| DM batch size | 64 | 128 | 128 |
| DM epochs | 10 | 10 | 10 |
| DM Hidden units | 64 | 128 | 128 |
| DM buffer size | 480 episodes | 640 episodes | 1280 episodes |
| DM prediction | $o_{t+1}$ | $o_{t+1}$ | $puck - observation_{t+1}$ |
| Truncated backpropagation through time (tbptt) | True | True | True |
| TBPTT window lentgh | 50 | 50 | 50 |
| predictability regularization $\alpha$ | 0.2 | 0.2 | 0.2 |
| predictability regularization $\beta$ | 0.1 | 0.1 | 0.1 |

### D.2   K-NN ENTROPY ESTIMATION

The original estimator proposed by Singh et al. (2003):

$$\hat{H}_N^k(Z) = \frac{1}{N} \sum_{i=1}^{N} \log \frac{N \cdot \|z_i - z_i^{k\text{-NN}}\|_2^{q/2} \cdot \hat{\pi}^{q/2}}{k \cdot \Gamma\left(\frac{q}{2} + 1\right)} + C_k \tag{7}$$

Where $z_i^{k\text{-NN}}$ is the k-NN of $z_i$ within a set $\{z_i\}_{i=1}^{N}$, $C_k = \log(k) - \Psi(k)$ is a bias correction term, $\Psi$ the digamma function, $\Gamma$ the gamma function, $q$ the dimension of $z$, $\hat{\pi} \approx 3.14159$. as mentioned before, this estimator is provably asymptotically unbiased and consistent.

**Intrinsic reward adaptation:** As mentioned before, Liu & Abbeel (2021b) were the first to optimize the entropy objective by breaking the K-nn entropy estimator into intrinsic rewards as in Equation 6. This approach has been later adapted by both off-policy (Seo et al., 2021; Yarats et al., 2021) and on-policy (Kim et al., 2023; Ashlag et al., 2025) algorithms. As for the numerical stability constant we used $c = 1$.

## E    EXPERIMENTAL DETAILS

Once again, code for both LatEnt and PROBE is available at `https://github.com/JonathanAshlag/LatEnt`.

**Pretraining baselines:** As stated in the main paper, to implement the observation entropy maximization baseline, we use the LatEnt algorithm and modify only the entropy computation to operate on observed states. While prior algorithms (Liu & Abbeel, 2021b; Yarats et al., 2021) could have been considered, we found no compelling reason to employ them as they differ primarily in two aspects:

- **Vision-based domains**: These works focus on maximizing observation entropy in high-dimensional image spaces and therefore emphasize learning compact observation representations to enable more accurate entropy estimation in lower-dimensional spaces. However, the PROBE observation space consists of proprioceptive vectors with no redundant dimensionality requiring compression.

- **Off-policy vs. on-policy entropy maximization**: it remains an open question whether to optimize entropy objectives using low-variance, biased off-policy approaches (Seo et al., 2021; Liu & Abbeel, 2021b; Yarats et al., 2021) or unbiased, higher-variance on-policy methods (Mutti et al., 2021). Since this paper adopts the latter approach, comparing against on-policy baselines allows us to isolate the effect of the objective from the optimization scheme.

**Finetuning details**: PPO: We adopted Cleanrl's Huang et al. (2022) continuous action PPO implementation to fit for recurrent policies. As mention in the main body, all baselines use the same architecture as LatEnt, a recurrent GRU based backbone (built the same as the latent dynamics model) with an MLP policy on top.

**Finetuning hyperparameters sweep**: For each PPO-baseline, including LatEnt, for each environment, we swept for 3 hyperparameters and took the best performing on average across the environments downstream tasks:

Batch size $\in [4, 8, 16]$, Clip coefficient $\in [0.2, 0.3]$, Action entropy coefficient $\in [0.003, 0.001]$

For PPO from scratch we additionally swept for Action entropy coefficient across $[0.03, 0.01, 0.005]$

DreamerV3 baseline: We used the widely adopted DreamerV3 Pytorch implementation available at https://github.com/NM512/dreamerv3-torch under the MIT license. For hyper-parameters we used the continuous control form vector observations (DMC-proprioceptive) configuration.

**Ablation Studies**: LatEnt with no predictability regularization: we set $\alpha, \beta = 0, 0$ in equation 5.

LatEnt with history encoding: Following Zintgraf et al. (2019) the dynamics model is trained to produce hidden states $\ell_t$ that can re-predict all past transitions:

$$\sum_{k=1}^{t+1} p_\theta(o_k | o_{k-1}, a_{k-1}, \ell_t)$$

Where we calculate the probability as Euclidean distance. Besides changing the prediction MLP from $\hat{o}_{t+1} = p_\theta(\ell_t, a_t)$ to $\hat{o}_{t+1} = p_\theta(\ell_t, a_t, o_t)$ we used the same architecture as LatEnt.

