# OpenReview forum: "Probing in the Dark: State Entropy Maximization for POMDPs"
_ICLR.cc/2026/Conference — ICLR 2026 Poster_

### Official Review · Reviewer_B9Ww · 2025-10-28

**Soundness:** 3
**Presentation:** 3
**Contribution:** 3
**Rating:** 6
**Confidence:** 3

**Summary:**

Authors propose a new reward-free RL pre-training entropic objective ("information state") applicable to POMDPs. They analyze this representation statistic on sufficiency and compactness. Moreover they evaluate (1) whether pre-training with LatEnt indeed boosts the initial state entropy, and (2) whether pre-training with LatEnt improves RL sample efficiency over two strong baselines. The answers are yes in three toy environments.

**Strengths:**

Originality and significance: The analysis on the sufficient/compact statistics is novel in the context of POMDP latent state representation. That said, I am not completely up-to-date on RL pre-training. Empirical results are strong.

Quality and clarity: The experiment design and ablation studies seem solid and precisely target RQs. One can always ask for more environments or more baselines, but I think in this case the theoretical contribution and empirical results stand as is.

**Weaknesses:**

One confounder is exploration frames. What if you give the baselines (ppo and dreamer) extra global step budgets for what they have missed out in pre-training? If they still cannot sample effectively, this result would strengthen your IS argument even more, beyond the basic "pre-trained policies had longer simulation time to explore".

**Questions:**

What if your objective is used for auxiliary regularization during RL instead of two-staged training? It has been done in MDP and could potentially reuse some rollouts.

---

> ### Author Response · Authors · 2025-11-20
>
> We thank the reviewer for their insightful comments. We are glad they appreciated our work and identified strengths both in terms of novelty and significance as well as quality and clarity, with the experiments deemed solid and precise. We reply to their questions below.
>
> **1. "One confounder is exploration frames. What if you give the baselines (ppo and dreamer) extra global step budgets for what they have missed out in pre-training?"**
>
> The aim of our research is to provide a method for pre-training a policy in POMDP without rewards. As it is common practice in previous works on policy pre-training (e.g., Mutti et al. 2021; Liu \& Abbeel 2021), the pre-trained policy is evaluated against standard methods that are learning from scratch.
>
> The reviewer rightly notice that PPO and Dreamer could also benefit from the additional steps used during pre-training, possibly outperforming LatEnt. Indeed, if the goal is to solve one task only, we do not advocate for any pre-training, but to run the best reward maximization algorithm instead (our objective may still be relevant as a regularizer, see below). However, a single policy pre-trained with LatEnt can be fine-tuned to solve several different tasks. The additional cost of pre-training fades away the more tasks we aim to solve with the pre-trained policy.
>
> **2. "What if your objective is used for auxiliary regularization during RL instead of two-staged training? It has been done in MDP and could potentially reuse some rollouts."**
>
> This is a good point! While the focus of this work is policy pre-training, for which the RL fine-tuning experiments are mostly meant to evaluate the pre-training, our latent entropy can provide further benefit as a regularizer. This has been shown to improve sample efficiency (e.g., Seo et al. 2021) and robustness (Ashlag et al. 2025) in MDP settings. We believe that similar benefits can be established for the POMDP setting we tackle here, which makes for a compelling direction for future investigations.

---

> > ### Comment · Reviewer_B9Ww · 2025-11-26
> >
> > Thank you for the responses and they help clarified my questions and concerns.

---

> > > ### Author Response · Authors · 2025-11-26
> > >
> > > Thank you for considering our response. We are happy to hear that the question about the exploration frames has been resolved. If all the concerns have been addressed, we kindly ask the reviewer to consider raising their score, which currently reads as "marginally above the acceptance threshold. But would not mind if paper is rejected"

---

### Official Review · Reviewer_LwEB · 2025-10-28

**Soundness:** 2
**Presentation:** 2
**Contribution:** 3
**Rating:** 4
**Confidence:** 4

**Summary:**

The work concerns unsupervised pretraining of (exploration) policies for partially observable environments. The work discusses in detail the difference between maximum state entropy and maximum observation entropy to motivate why the former is more informative. Still as it this would require knowledge of true states, the work introduces a surrogate in the form maximum latent entropy as objective. In this setting the objective is to maximize the entropy over the latent states that are induced by the current policy under the current dynamics model. The work then discusses all relevant design decisions to derive the LatEnt algorithm for on-policy pretraining using PPO.
To assess the quality of the proposed method the work introduces the PROBE benchmark, which provides partially observable variants of existing benchmarks, such as Pendulum or Ant. The work empirically verifies that the maximum latent entropy objective outperforms the maximum observation entropy objective.

**Strengths:**

The work tackles an important problem by proposing a novel solution. I am a big fan of code being available already during the review phase and want to highlight it. The work does a good job discussing the different pretraining objectives and explaining the proposed solution approach and positioning itself in the related research. I only have minor questions related to the approach:
* Isn't the choice of null-action for $a_{-1}$ (as stated on line 228) potentially very environment specific? Would it make sense to sample this value from the full action range to enable potential different initializations?
* Is the choice of PPO for policy learning crucial or is LatEnt agnostic to the policy learning mechanism, as long as dynamics model learning updates happen less frequently than policy learning? I am wondering if the constraint updates of PPO are particularly benefitial for the way LatEnt is learning the representations.
* Lines 295- 297 claims that a larger batch size is required for the entropy objective but it's not substantiated how much bigger this is required and I do not immediately see the reasoning behind this statement.

Minor comments:
* Line 237 missing space after colon symbol
* Footnote 1: Shouldn't the reward still be part of the POMDP definition, even if you consider the reward free setting or differing tasks?

**Weaknesses:**

The introduction of the PROBE benchmark seems like the biggest weakness to me. I) The environment descriptions are confusing. For example it is not fully clear from the text what exactly it means that something is unobservable in Pendulum if $y=0$ (line 319). It is also not well motivated why Pendulum requires both an unobservable region and the occlusion of the velocity from the observation vector. Similarly, the statement that Ant's z-axis is fully observable only when the ant is on the ground (line 337) requires a bit more explanation. Does that mean that at least one of the legs needs to touch the ground or is there the same cutoff as in Pendulum? (Fiugre 1 b suggests the latter). Directly after, it is stated that extrinsic forces are unobserved at all times in Ant but it is not clarified what exactly that means. I intuitively understood it as a form of "random wind" being applied to the environment that is not directly observed by the agent but has to be inferred. Lastly in Pusher it is not clear if the noisy puck position is just on the observation or if it is noise on the true state.
These details need to be clarified to better justify the choice of introducing and using these benchmarks.
I understand that there are more descriptions of the environments in Appendix C, but those do not provide descriptions of why the environment modifications were chosen. In any case, it is unlikely that most readers will immediately look into Appendix C when reading the paper and would be more confused by the unclear environment description of the main paper.

Further, I understand that the authors feel that existing POMDP benchmarks are inadequate for their purposes (Line 308-309). Still, I believe it would be good to evaluate LatEnt on at least one such benchmarks to better quantify the advantages/disatvantages that stem from the novel algorithm. Solely evaluating on a newly proposed benchmark seems questionable to me.

As it stands, I believe this is a very interesting paper with a lot of interesting ideas that should spark good discussions at the conference. However, the above shortcomings in experimental design make me hesitant to vote for acceptance. In particular the sole focus on the newly introduced environments seems indefensible to me. For now I vote borderline leaning towards rejection. I am happy to increase my score if the authors provide a better justification why only the PROBE benchmark results should be deemed sufficient or provide some small-ish experiments that show LatEnts behavior on established benchmarks.

**Questions:**

Please see the sections above.

---

> ### Author Response · Authors · 2025-11-20
>
> We thank the reviewer for the valuable comments. We are happy to hear they found the method a novel solution to an important problem, while the paper provides ``a lot of interesting ideas that should spark good discussion at the conference''.
>
> We hear their concerns about the PROBE benchmark. We provide extensive clarifications on why a new benchmark is needed in the **general response**.
> We hope to convince the reviewer why we think PROBE is a strength of our work rather than a weakness, as it addresses an important gap in the literature when it comes to evaluating POMDP methods.
>
> **PROBE benchmark.**
>
> **"The environment description are confusing."**
>
> We answer to the reviewer points below, we aim to make use of this useful feedback to improve the environment description in the paper and the appendix. we will post a modified version in the next week.
>
> Technical descriptions.
> - Pendulum: When the pendulum is in the upper half of the circle, the true position of the pendulum is masked. The velocity dimension is always masked. Notice that the latter is not *required* for the domain to be partially observable. It is a design choice to make the problem harder.
> - Ant: The cutoff is applied to the $z$-coordinate of the torso. External forces (78 element of the observation vector) are the ``center of mass based external forces on the body parts'' (documentation). All of those are masked out in our implementation.
> - Pusher: Only the observation is noisy, there is no noise applied to the true state.
>
> **1. "Provide a better justification why only the PROBE benchmark results should be deemed sufficient."**
>
> Thank you for raising this important point. Given its importance, we provided an extensive clarification in the **general response**.
>
> **2. "It would be good to evaluate LatEnt on at least one existing benchmarks to better quantify the advantages/disadvantages that stem from the novel algorithm."**
>
> At the request of the reviewer, we run an additional experiment in the DeepMind Control suite (Tassa et al. 2018), which is the common benchmark in prior works (e.g., Seo et al. 2021, Yarats et al 2021). We summarize the results in the **general response**.
>
> **3. Choice of the null action in the first step**
>
> The choice of the null action is standard in previous work (e.g., Zhu et al "On improving deep reinforcement learning for pomdps" 2017, Igl et al "Deep Variational reinforcement learning for pomdps" 2018). We believe the strategy of taking a random action from the action range, as suggested by the reviewer, is also valid. We do not expect any impact on the reported results.
>
> **4. Choice of PPO for policy learning**
>
> This is a very good point that we are happy to clarify. PPO is not a crucial component of the algorithm, although we agree that the strengths of PPO mesh well with the implementation of the latent model. To demonstrate this, we reimplemented LatEnt with A2C and evaluated it on Delusional Pusher. A2C-LatEnt achieved mean state entropy of 0.082 $\pm$
>  0.03 (5 seeds), which is 60\% of PPO-LatEnt's performance. Notably, A2C-LatEnt requires careful hyperparameter tuning, only working for a narrow range of learning rates and update frequencies, whereas PPO is stable across a broader range of values. %, highlighting how PPO's robustness complements our approach.
>  Converting our approach to use off-policy RL algorithm is less trivial. We leave this direction as future work.
>
> **5. Larger batch size?**
>
> The batch size is domain dependent and it is not straightforward to tell how large shall it be in general. However, we note that the required batch size will typically be larger than what is needed for standard PPO. This is because we are estimating the gradient of an estimated function (the entropy), instead of estimating the gradient of an observed quantity (the sum of the rewards). The entropy estimation is an additional source of statistical complexity, formally analyzed in Singh \& Poczos 2016. Ashlag et al. 2025 provides a more informal and practical analysis of this effect.

---

> > ### Comment · Reviewer_LwEB · 2025-11-20
> >
> > Thank you for the clarifications. I have by now read all other reviews and the responses.
> > I am a bit more positive towards the novel benchmark, though I am not fully convinced yet. Could the authors maybe map the design choices in the individual environments affect the emission matrix as well as the Hadamard inverse as stated in the general comment?
> > In the meantime I will increase my score to borderline accept

---

> > > ### Author Response · Authors · 2025-11-23
> > >
> > > Thank you for the follow up on our replies and for reconsidering your stance towards the PROBE benchmark.
> > >
> > > Mapping how the design choice affect the properties of the observation function is actually a great idea.
> > >
> > > **Masked Pendulum.**
> > > Since the domain is low-dimensional we can run a small computational experiment to show the effect of our changes. We aim to show that our changes result in the $\sigma_{\text{max}} (O)$ to increase. We discretize the pendulum position and velocities in a grid 16x16 (256 states in total).
> > >
> > > - The original implementation has $\sigma_{\text{max}} (O) = 1$
> > >
> > > - Masking the position in the upper half of the circle $\sigma_{\text{max}} (O) \approx 2.82$
> > >
> > > - Masking the position in the upper half of the circle and velocities everywhere $\sigma_{\text{max}} (O) \approx 11.313$
> > >
> > > This analysis backs up our design choice to mask both the position and velocities.
> > >
> > > **Vertically Blind Ant.** Since the problem is higher dimensional, computing the observation matrix of the discretized space is cumbersome in ant. In words, masking a certain region of the state space will make all of the rows of the emission matrix related to that region to collapse to the same probability distribution over observations. The resulting matrix changes from an identity matrix to a matrix with several equivalent rows, which makes the $\sigma_{\text{max}} (O)$ increase.
> > >
> > > **Delusional Pusher.** In the pusher domain we apply a Gaussian noise to the observation of the puck position. This makes the $\sigma_{\text{max}}(O^{\circ - 1} )$ increase without changing the $\sigma_{\text{max}}(O )$. However, since the Gaussian noise is dependent on the distance between the puck and the end effector, the $\sigma_{\text{max}}(O )$ shall also increase slightly. This double effect of the noise makes the pusher domain the hardest in the benchmark, as captured by our results.

---

> > > > ### Comment · Reviewer_LwEB · 2025-11-24
> > > >
> > > > Thank you very much for your response. This highlights the utility of the benchmark much better and I am happy to increase my score again. I would appreciate if this discussion would be added to the paper (at least the appendix).

---

> > > > > ### Author Response · Authors · 2025-11-26
> > > > >
> > > > > Thank you for your constructive feedback, which give us the chance to improve the paper by clarifying the value of the benchmark. We are adding the discussion to the manuscript (the main points to the main text and technical details to the appendix). We will upload the updated version of the paper soon.

---

### Official Review · Reviewer_cjki · 2025-10-31

**Soundness:** 2
**Presentation:** 1
**Contribution:** 2
**Rating:** 2
**Confidence:** 4

**Summary:**

This paper extends maximum entropy unsupervised RL for MDPs to POMDPs. It formulates a new training objective and proposes a training algorithm called LatEnt, which is evaluated on a new benchmark.

**Strengths:**

* Pretraining a policy for POMDP to improve sample efficiency is an important problem
* The paper is well-organized at a high level.

**Weaknesses:**

I find this paper's work insufficiently or misleadingly motivated. The popular state entropy maximization approach for pretraining or encouraging exploration for MDPs has previously been extended to POMDPs, as cited by the authors, which shows interest in doing such extension. When it comes to the paper's work, there was just a brief reference to Zamboni et al. (2024b) that "these approaches are bound to fail on POMDPs with more general observability properties", but there is no discussion and example of what the "general observability properties" could be, and why these cases are important; in addition, Zamboni et al. (2024a) is one of "these approaches", but as far as I can see, Zamboni et al. (2024b)'s analysis is not applicable to this work. Furthermore, Zamboni et al. (2024a) actually does maximize state entropy, instead of observation entropy.

Another major concern I have is that the technical writing lacks sufficient clarity. I'll mention some vague definitions, claims and theorems below.
* Definition 1: What does it mean to say that IS2 can be replaced by IS2a and IS2b? Presumably doing this leads to a different notion of information state, but the wording is confusing.
* Theorem 1: This theorem shows that the concept of information state in Definition 1 is the "an information state for a POMDP with a convex objective", but the latter has not been defined yet.
* Assumption 1: I find it rather confusing to assume that it is possible to access the state of a POMDP for a constant number of times.  If it is possible to access the state of a POMDP, then it is not a POMDP any more, at least not in the standard sense. In addition, I find the claim "Assumption 1 rules out the possibility to maximize the state entropy directly" vague and hand-wavy.
* Definition 2: $\ell$ is not used in IS2a and IS2b.

Some of these may be easily fixable, but with a general lack of precision in technical writing, it is very difficult to assess the correctness of the claims.

The proposed benchmarks appear to be somewhat contrived, as there is no discussion on what kind of "general observability properties" they are designed to capture.

Also, is there a reason why there is no empirical comparison with Zamboni et al.  (2024a)? And some other more recent approaches as cited in Zamboni et al. (2024b)?

**Questions:**

Please refer to weaknesses.

---

> ### Author Response · Authors · 2025-11-20
>
> We thank the reviewer for their useful comments and for pointing out sources of confusion. We hope that our clarifications below will make them better appreciate the contribution of the paper.
>
> **1. Work is insufficiently or misleadingly motivated[...] Related works Zamboni et al. 2024ab.**
>
> We thank the reviewer for pointing this out. We are very familiar with the work of Zamboni et al. 2024ab but we acknowledge it is not straightforward to understand how we advance over them from the manuscript alone. We will introduce some changes in the text to make the paper more self-contained in the next week. We summarize the key points below.
>
>
> Zamboni et al. 2024b address the exact same problem formulation of our paper (see Eq. 2). They analyze the same maximum observation entropy objective we consider (Eq. 4) and compare to in the experiments. Their main result (Th. 4.1) shows that  maximizing the observation entropy is enough under some conditions (see the \textbf{general response}). In this paper, we tackle settings in which those conditions do not hold, which is left as future work in Zamboni et al. 2024b (see their Conclusion section).
>
> Zamboni et al. 2024a is also targeting this open direction as we do. Their proposed solution is to maximize the entropy of the distribution of states sampled from the belief. This has some similarity with our latent entropy objective. There is an important caveat, however: Zamboni et al. 2024a assume the belief to be given. In practice, the belief has to be estimated from data. Moreover, their approach is only tested in small gridworld domains with finite states and observations. How to scale their approach to approximate beliefs and continuous settings is left as future work (see their Conclusion section).
>
> Our paper tackles both the open problems left by Zamboni et al. 2024ab by providing a scalable algorithm for state entropy maximization in continuous domains, without assuming given belief or small $\sigma_{\text{max}} (O)$ and $\sigma_{\text{max}} (O^{\circ - 1})$. We hope this discussion clarifies the motivation behind our work and how it substantially advances over the literature.
>
> **2. "Technical writing lacks sufficient clarity"**
>
> We thank the reviewer for pointing out the issues. We provide clarifications below, we will update the manuscript accordingly in the next week. We are available to provide additional clarifications on any remaining concern.
>
> - **Definition 1:** It means that IS2a and IS2b imply IS2. Since IS1 + IS2 are sufficient conditions for an information state, then also IS1 + IS2a + IS2b are. It is the same notion of information state, just slightly more demanding sufficient conditions.
>
> - **Theorem 1:** The standard objective of a POMDP, i.e., reward maximization, is linear in $d^\pi (s)$. We refer to a POMDP with a convex objective a POMDP having an objective that is a convex/concave function of $d^\pi (s)$, including, but not limited to, the entropy. This terminology is common in MDPs (e.g., Zahavy et al 2021 "Reward is enough for convex mdps").
>
> - **Assumption 1:** We want to clarify the assumption says *we can access the true states a number of times that is upper bounded by a constant*.
>     Our algorithm does not use the true states (Alg.1), but notice it is impossible to evaluate its entropy on the true states without some access to them. Thus, in our experiments a small batch of true states is used only for evaluation of different hyper-parameters. Finally, the claim that constant access of the true states does not allow for learning is made formal in the provided reference (Lee et al. 2023): Learning requires polynomial access, constant access is not enough. We will clarify in the text that this is a formal result from prior work and not our claim.
>
> - **Definition 2:** $l$ takes the place of $z$ and $L$ incorporates the functions $\sigma_t$ and $\phi_t$.
>
>
> **3. "The proposed benchmarks appear to be somewhat contrived, as there is no discussion on what kind of "general observability properties" they are designed to capture."**
>
> We thank the reviewer for pointing this out, which allows us to address an important source of confusion (see the **general response**).
>
> **4. "Also, is there a reason why there is no empirical comparison with Zamboni et al. (2024a)? And some other more recent approaches as cited in Zamboni et al. (2024b)?"**
>
> As discussed above, Zamboni et al. 2024a do not provide a practical algorithm we can compare to. Their approach assume access to the belief and is limited to discrete settings. We are not aware of other approaches targeting maximum state entropy in POMDPs we may compare to.

---

### Official Review · Reviewer_jeJm · 2025-10-31

**Soundness:** 3
**Presentation:** 2
**Contribution:** 3
**Rating:** 4
**Confidence:** 3

**Summary:**

The paper introduces an exploration technique in POMDPs. It consists of maximizing the frequency with which an information state of the POMDP is visited.

**Strengths:**

1. Exploration in POMDPs is a complex problem that, to my understanding of the literature, has been little studied.
2. The proposed method is intuitive and seems correct.
3. The experiments are comprehensive and clear.

**Weaknesses:**

1. The presentation is rather heavy, and the paper is difficult to follow.
2. From what I understand, there is an incompleteness in the formalization. The distribution $d^\pi_L(l)$ is never explicitly defined. However, this object is at the heart of the method. I think it is important to clarify the definition of this object.
3. From my understanding of section 4.1, a deterministic latent model is used and learned by minimizing the mean squared error (L2 reconstruction). If this is indeed the case, we are trying to learn a very specific type of information state that is "deterministic" predictive, where the definition of Subramanian et al. (2022) only required "stochastic" predictiveness. From my understanding, such IS only exists if the history allows the state to be reconstructed deterministically, which is a very specific case of POMDP (like memory POMDP).

**Questions:**

1. Could the authors clarify what $d^\pi_L(l)$ is? Could authors also clarify what it would equal in the particular case of using the belief as IS?
2. Could the authors clarify if the latent space is deterministic or not? If it is, what guarantee do we have that the learned statistic is indeed an IS?

---

> ### Author Response · Authors · 2025-11-20
>
> We thank the reviewer for their insightful comments. We are glad they think our paper is targeting a complex, understudied problem, for praising the proposed method, and finding the experiments clear and comprehensive.
>
> **Q1. Could the authors clarify what $d^\pi_L$ is? Could authors also clarify what it would equal in the particular case of using the belief as IS?**
>
> $d^\pi_L$ is the distribution over latent states induced by the model $L$ and the policy $\pi$. Formally, $d^\pi_L (l) = \sum_{t \in [T]} \mathbb{P} (l_t = l | \pi, L) / T$. Intuitively, $d^\pi_L (l)$ captures the probability of visiting a latent state $l$ within a $T$-steps trajectory taken with the policy $\pi$. This matches the definition of $d^\pi (s)$ (line 132) but for latent states instead of true states. When the latent space is continuous, $d^\pi_L (l)$ denotes the probability density function.
> The belief is a special case of predictive latent, so it fits the same definition.
>
> We will include the formal definition of $d^\pi_L (l)$ in the paper to avoid this source of confusion. A new version incorporating this clarification will be available within a week.
>
> **Q2. Could the authors clarify if the latent space is deterministic or not? If it is, what guarantee do we have that the learned statistic is indeed an IS?**
>
> We are not sure what do they mean by ``deterministic latent space''.
>
> If the reviewer is concerned that the prediction of the next observation, given by $p_\theta (\ell_t, a_t)$, is deterministic: Learning $p_\theta$ by minimizing the MSE implicitly assumes that the observation given the latent is distributed as a Gaussian. This works nicely in our continuous control experiments. More expressive distribution can be considered with little change to the algorithm. To demonstrate this flexibility, we replaced the deterministic decoder with a VAE. With minimal hyperparameter tuning (KL coefficient = 0.01), LatEnt-VAE achieved improved performance on both tested environments (Table below, 10 seeds, mean and std):
>
> **Table: State Entropy Achieved in Pretraining**
>
> |                              | **Vertically Blind Ant** | **Delusional Pusher** |
> |------------------------------|---------------------------|-------------------------|
> | **Max state entropy**        | 0.93 ± 0.022              | 0.035 ± 0.001           |
> | **LatEnt**                   | 0.85 ± 0.014              | 0.013 ± 0.003           |
> | **LatEnt-VAE**               | 0.86 ± 0.015              | 0.016 ± 0.006           |
>
> If the reviewer is concerned that the evolution of the latent state, controlled by $f_\theta (o_t, a_{t - 1}, \ell_{t - 1})$, is deterministic: This is without loss of generality, there always exists an information state in this class, e.g., the history and the belief both evolve deterministically given the action and next observation.

---

### Author Response · Authors · 2025-11-20
**Clarification on the ``general observability properties'' and the PROBE benchmark**

Reading reviewers' comments, we noticed a source of confusion deriving from an insufficient elaboration on what we mean by ``general observability properties'', why current benchmarks do not have them, and why we designed a new benchmark with those properties.

In the paper, we mention the ``general observability properties'' we aim to tackle with a reference to Zamboni et al 2024b. Their main result (Th. 4.1) shows that  maximizing the observation entropy is enough whenever the maximum singular value of the emission matrix $\sigma_{\text{max}} (O)$ and the Hadamard inverse (having elements $O_{ij}^{\circ -1} = 1 / O_{ij}$) of the emission matrix $\sigma_{\text{max}} (O^{\circ - 1})$ are both small. Coarsely, $\sigma_{\text{max}} (O)$ is large when a state can emit various distinct observations, while $\sigma_{\text{max}} (O^{\circ - 1})$ is large when a single observation can be emitted by several distinct states. In our paper, we aim to target domains where either $\sigma_{\text{max}} (O)$ or $\sigma_{\text{max}} (O^{\circ - 1})$ is large.

The most common benchmark in prior works, the DeepMind Control suite (Tassa et al 2018), $\sigma_{\text{max}} (O^{\circ - 1})$ and $\sigma_{\text{max}} (O)$ are both relatively small. $\sigma_{\text{max}} (O)$ is small because every state can emit just one observation. $\sigma_{\text{max}} (O^{\circ - 1})$ is not necessarily small as every observation can be emitted by several distinct states (e.g., same position with different velocities), but it becomes small once you stack a few consecutive observations (frames), which is the default setup of prior works. In this kind of settings, Zamboni et al. 2024b shows that maximizing the observation entropy is enough.

In our paper, we aim to target settings in which either $\sigma_{\text{max}} (O)$ or $\sigma_{\text{max}} (O^{\circ - 1})$ is not small, which we informally describe as ``general observation properties''. To this end, we develop a new benchmark with those properties: In *Delusional Pusher* every state can emit several distinct observations (large $\sigma_{\text{max}} (O)$), in *Masked Pendulum* and *Vertically Blind Ant* the observations of the masked regions are emitted by multiple states (large $\sigma_{\text{max}} (O^{\circ - 1})$).

Finally, to address the concern that the proposed method is only evaluated in a new benchmark, we conducted additional experiments in the most common DeepMind Control suite (Tassa et al. 2018).
We summarize in the table below the results obtained by LatEnt against the maximum observation entropy (solution proposed by prior works) and the maximum state entropy (ideal target). As the reviewers can see, LatEnt and maximum observation entropy achieve nearly identical performance across all metrics. This is in line we the explanation above and the results of Zamboni et al. 2024b.

**Table: DMC Walker pretraining (10 seeds, mean ± std)**

|                              | **State entropy**     | **Observation entropy** | **Latent entropy**    |
|------------------------------|------------------------|--------------------------|-------------------------|
| **Max state entropy**        | 2.61 ± 0.008          | 1.528 ± 0.011           | 0.847 ± 0.013          |
| **Max observation entropy**  | 2.35 ± 0.027          | 1.589 ± 0.012           | 0.85 ± 0.01             |
| **LatEnt**                   | 2.36 ± 0.012          | 1.58 ± 0.005            | 0.863 ± 0.012           |
| **Random policy (baseline)** | 2.26 ± 0.001          | 1.49 ± 0.001            | NA                      |


**Table: DMC Hopper pretraining (10 seeds, mean ± std)**

|                              | **State entropy**     | **Observation entropy** | **Latent entropy**    |
|------------------------------|------------------------|--------------------------|-------------------------|
| **Max state entropy**        | 1.77 ± 0.056          | 1.19 ± 0.039            | 0.757 ± 0.042          |
| **Max observation entropy**  | 1.73 ± 0.03           | 1.3 ± 0.01              | 0.81 ± 0.01            |
| **LatEnt**                   | 1.71 ± 0.03           | 1.28 ± 0.024            | 0.818 ± 0.017          |
| **Random policy (baseline)** | 1.39 ± 0.002          | 1.04 ± 0.001            | NA                     |

---

### Author Response · Authors · 2025-11-28
**Summary of Changes in the Revised Version**

Dear reviewers,
Following your comments and suggestions, we have uploaded a revised version of the paper. We believe the clarity of the contribution has improved.
The main changes, which are marked in blue, are summarized as follows:

- In the PROBE subsection, we added a discussion on Zamboni et al. (2024b), including the meaning of the bound, $\sigma_{\text{max}}(O^{\circ -1})$ and $\sigma_{\text{max}}(O)$, and how they motivate our design choices for PROBE.
- The description of PROBE has been refined in both the main paper and the appendix.
- The DeepMind Control pretraining experiments have been added to the appendix.
- Notation in Sections 2–3 has been clarified.
- We have improved the discussion of our paper’s relationship to Zamboni et al. (2024a, 2024b), (Section 6).
- Section 4 now discusses how LatEnt could be implemented with more expressive predictive architectures (e.g., VAEs).

Best Regards,

The Authors

---

### Meta-Review · Area_Chair_QHWu · 2025-12-26

**Summary:**

The reviewers' main concern was the validity of the empirical evaluation, due to the lack of clarity regarding the environments and the motivation for the proposed method. In particular, it was not clear what "general observability properties" were and how the environment used in the empirical evaluation would support the claims that the proposed method would address such a problem.
The rebuttal was sufficient to clarify these concerns: the proposed approach can handle a more general case than the method from Zamboni et al. 2024b, which is also supported by the empirical evaluation once the details about the PROBE benchmark were clarified in the rebuttal.

**Reviewer Concerns:**

`cjki` was concerned regarding the motivation for the proposed method. In particular, it was not clear what "general observability properties" were. The rebuttal was sufficient to clarify this concern.

`jeJm` had two questions which were well addressed in the rebuttal. Furthermore, the authors included an extra experiment with a VAE, which improved the proposed method.

`LwEB` seemed very positive about the paper; however, they identified key issues (unclear environment choice, missing comparison with previous environments) that would prevent accepting the paper. The rebuttal fairly well tackled these issues.

`B9Ww` had some questions regarding the potential applicability of the proposal in an online setting (without pre-training), but no specific concerns about the proposed method. The authors addressed the question.

**Reviewer Scores:**

- `jeJm`: 4 -> 6
- `cjki`: 2 -> 4
- `LwEB`4 -> 8
- `B9Ww` 6 -> 6

---

### Decision · Program_Chairs · 2026-01-26

Accept (Poster)